# Protective Role of Kelulut Honey against Toxicity Effects of Polystyrene Microplastics on Morphology, Hormones, and Sex Steroid Receptor Expression in the Uterus of Rats

**DOI:** 10.3390/toxics11040324

**Published:** 2023-03-29

**Authors:** Nur Hanisah Amran, Siti Sarah Mohamad Zaid, Goh Yong Meng, Annas Salleh, Mohd Helmy Mokhtar

**Affiliations:** 1Department of Environment, Faculty of Forestry and Environment, Universiti Putra Malaysia (UPM), Serdang 43400, Selangor, Malaysia; 2Department of Veterinary Pre-Clinical Science, Faculty of Veterinary Medicine, Universiti Putra Malaysia (UPM), Serdang 43400, Selangor, Malaysia; 3Department of Veterinary Laboratory Diagnostic, Faculty of Veterinary Medicine, Universiti Putra Malaysia (UPM), Serdang 43400, Selangor, Malaysia; 4Department of Physiology, Faculty of Medicine, Universiti Kebangsaan Malaysia, Kuala Lumpur 56000, Malaysia

**Keywords:** microplastics, polystyrene, toxicity, uterus, reproductive, female, Kelulut honey

## Abstract

Background: Microplastics (MPs) are an emerging global pollutant. Previous studies have revealed that chronic exposure to MPs can affect animal and human reproductive health, particularly by impairing the reproductive system’s normal functions, which may increase the risk of infertility in both males and females. Kelulut honey (KH), an excellent source of antioxidants, has been used to counteract the disruptive effects of Polystyrene microplastics (PS-MPs) in the rat uterus. Thus, this study aimed to investigate the potential protective effects of Kelulut honey against PS-MPs-induced uterine toxicity in pubertal rats. Methods: Prepubertal female Sprague Dawley rats were divided into four groups (n = 8): (i) normal control group (NC: treated with deionized water), MPs-exposed group (M: exposed to PS-MPs at 2.5 mg/kg), (iii) Kelulut honey group (DM: pretreated with 1200 mg/kg of KH 30 minutes before they were administered with PS-MPs at 2.5 mg/kg), and (iv) Kelulut honey control group (DC: only treated with KH at 2.5 mg/kg). The rats were treated orally once daily for six consecutive weeks. Results: Uterine abnormalities in PS-MPs-exposed rats were significantly improved after concurrent treatment with Kelulut honey. Morphology improvement was observed and luminal epithelial cells seemed thicker with more goblet cells, glandular cells had a more regular and circular shape, stromal cell increased in size, interstitial gaps between stromal cells expanded, and the myometrium layer was thicker. Kelulut honey treatment also effectively normalized the suppressive effect of PS-MPs on the expression and distribution of sex steroid receptors (ERα and ERβ), as well as the level of serum gonadotropin (LH and FSH) and sex steroid (estradiol and progesterone) hormones. Conclusion: Kelulut honey can protect the female reproductive system against the disruptive effects of PS-MPs. The phytochemical properties of Kelulut honey might be responsible for these beneficial benefits. However, future studies are warranted to identify the mechanisms involved.

## 1. Introduction

Over the last century, plastic residues have dramatically increased and polluted our environment, both in terrestrial and aquatic ecosystems. These materials have been mass produced since the 1950s and have steadily increased over the years, with 33 billion tons expected by 2050. More seriously, plastics that remain in the environment may never fully degrade but break down into smaller particles known as microplastics (MPs). MPs are a new type of environmental pollutant with a diameter less than 5 mm. In 2015, the United Nations Conference on the Environment declared that MPs pollution is the second-most important scientific problem in environmental and ecological research [1].

There are two main sources of MPs: a primary source that originates from manufactured products containing plastic particles or powder, such as cosmetics, toothpaste, facial and body scrubs, lotions, detergents, and drug carriers, and a secondary source that comes from the degradation of larger pieces of plastics by mechanical abrasion or natural forces such as wave action, ultraviolet radiation, or biological degradation in the environment. The are harmful not only because of their chemical content, which includes numerous hazardous additives and residual monomers such as bisphenol A, toluene-diisocyanate, styrene, and nonylphenol, but MPs also exert their harmful effects by facilitating the transport of other toxic compounds such as heavy metals and persistent organic pollutants (POPs) into organisms [2]. Additionally, MPs can affect human health through the plastic-contaminated environment or food chain through the consumption of plastic-contaminated food sources such as fish, crustaceans, chicken gizzards, and even sea salt, as well as through food packaging, taps, and water bottles [3,4,5,6].

In aquatic animals such as Daphnia, Hydra attenuates [7], Medaka fish [8], and oysters [9], MPs have been found to have a reproductive detrimental effect. In mammals, some studies have shown that MPs cause lower sperm quality, disturbed hormone levels, and oxidative damage in the testes [10,11,12]. Based on a previous study, MPs have also been linked to increased oxidative stress and inflammation in the female reproductive system, which may contribute to morphological and functional problems [13]. Zhaolan et al., 2021 [14], also discovered that PS-MPs can impair the reproductive function of both male and female mice, leading to reduced fertility. Surprisingly, female mice exposed to PS-MPs had a much stronger effect on reproduction and fertility than male mice. In addition to their xeno-estrogenic properties, the adverse effects of MPs effects may be related to the formation of ROS that contributes to pathological disorder of female reproduction [15]. Other studies have found that PS-MPs can induce rat ovarian granulosa cells to undergo pyroptosis and apoptosis via the NLRP3/Caspase-1 signaling pathway. These findings showed that the administration of MPs can lead to reproductive organ dysfunction [16].

Kelulut honey is a multi-floral stingless bee honey from *Trigona* spp. and has higher antioxidant and antimicrobial properties than other honey (such as Tualang and Gelam honey) [17]. Previous studies have shown that Kelulut honey possesses pharmacological effects such as anti-inflammatory [18], wound healing [19], antioxidant [17,20], and anti-aging abilities [21], with the antioxidant abilities of Kelulut honey also thought to protect against oxidative stress. Kelulut honey has these health benefits because it contains active phenolic acids, flavonoids, enzymes, ascorbic acid, proteins, and carotenoid groups. Since honey is well-documented as a natural product against toxicity effects in the reproductive system [13,22,23,24,25], this study investigated the protective roles of Kelulut honey against the toxicity effects of PS-MPs.

While most research on the toxic effects of MPs has focused on aquatic organisms, studies documenting the potential health risk of MPs to the reproductive system, particularly in female mammals, are lacking. Hence, data on the effects of MPs on the female reproductive system using mammalian models would be needed for the human health risk assessment of MPs. To address this lack of information, this present study comprehensively and systematically investigated the toxic effects of polystyrene microplastics (PS-MPs) on the female reproductive system of rats. The results of this study provide new insights and some primary data on the potential health risk of PS-MPs exposure to human health. This study also provides new important scientific information to support a rational intake of natural products in daily life to prevent further reproductive toxicity due to PS-MPs exposure.

## 2. Materials and Methods

### 2.1. Kelulut Honey (Agromas, Malaysia)

Kelulut honey was obtained from the Federal Agricultural Marketing Authority (FAMA) in Kuala Nerang (Kedah), which is under the Malaysia Agriculture Research and Development Institute (MARDI), Malaysia. Kelulut honey is polyfloral honey collected from Heterotrigona itama’s beehive in the farm that is built horizontally in hollow trunks, underground, and in wall cavities. The honey was processed through several stages at Honey Processing Centre in Kuala Nerang, Kedah (quality inspection, dehydration, packaging, and labeling). The honey was filtered to remove solid particles, concentrated in an oven at 40 °C, and subjected to *γ* irradiation at 25 kGy at Sterilgamma (M) Sdn. Bhd. (Selangor, Malaysia). The water content of the honey was standardized to 18% by FAMA.

### 2.2. Animal Model and Experimental Design

28-day old female Sprague Dawley rats were obtained from the UPM Animal Resource Unit, Faculty of Veterinary Medicine, University Putra Malaysia. All experimental designs and methods were conducted according to the National Institutes of Health Protocol for the Care and Use of Laboratory Animals (NIH Publications number 8023, amended 1978), which was approved by the Institutional Animal Care and Committee (IACUC) of the Universiti Putra Malaysia with the ethics reference code: UPM/IACUC/AUP-R081/2018. All animals were maintained under standard laboratory conditions at a temperature of 25 °C ± 2 °C and relative humidity of 50% ± 15% with a standard cycle of 12 h dark and 12 h light cycle and fed ad libitum with commercial pellet diet (Gold Coin Feedmills Pte. Ltd., Kuala Lumpur, Malaysia). To minimize exposure to endocrine disruptors or plastics, animals were housed in stainless steel cages with wooden bedding, and water was provided in glass bottles. The 32 female rats were randomly divided into four groups (*n* = 8 in each group) as follows:(1)Normal control group (NC), treated with vehicle (0.2 mL deionized water);(2)Microplastic control group (M), treated with polystyrene microplastic (PS-MPs) at 2.5 mg/kg body weight;(3)Kelulut honey dose + MPs group (DM), pretreated with 1200 mg/kg Kelulut honey for 30 minutes before administrated of PS-MPs at 2.5 mg/kg body weight;(4)Kelulut honey control group (DC), treated with 1200 mg/kg body weight of Kelulut honey only.

Administration by oral gavage was performed once daily in the morning (between 9:00 a.m. and 10:00 a.m.) for six consecutive weeks. The weekly body weight of each rat was recorded. At the end of the administration period, the rats were sacrificed in the diestrous phase. Blood samples were collected from the abdominal aorta under general anesthesia and the extracted serum was kept at −20 °C until further analysis.

The wet weight of the whole uterus was measured. The left horn of the uterus was fixed in 10% formalin for further histopathological and immunohistological analysis while the other half was kept in RNA for mRNA extraction. The dose selection of PS-MPs at 5 mg/kg body weight was based on a previous study in which PS-MPs at this dose disrupted the morphological and biochemical parameters in the reproductive system [26]. The Kelulut honey was freshly prepared daily to avoid oxidation of the antioxidant content. The dose of Kelulut honey at 1200 mg/kg body weight was based on a previous study stating that this dose is the most effective against toxicity on female reproductive organs [23].

### 2.3. Histopathological and Histomorphometric Analysis of the Uterus

Paraffin-embedded tissues were cut into 5 µm thick sections and then stained with hematoxylin-eosin according to standard protocols from a previous study [27]. For histomorphometric analysis, the slides were reviewed, and the clearest section on each slide was photographed at 10×, 20×, and 40× magnification. The mean values of the height of the luminal epithelial cells, the thickness of the endometrial and myometrial layers, the height of the endometrial epithelial glands, and the diameter of the endometrial glands were measured on six randomly selected areas of the sections. The number of endometrial glands was also counted. Histomorphometry was analyzed using a computer image analysis program (NIS -Elements Advanced Research, Nikon, Japan).

### 2.4. Assessment of Serum Estrogen, Progesterone, Follicle-Stimulating Hormones (FSHs), and Luteinizing Hormones (LHs) Levels

After anesthesia, blood samples were collected by cardiac puncture and centrifuged at 4 °C at a speed of 2500 rpm, to extract the serum. Serum levels of 17β-estradiol, progesterone, LH, and FSH were assessed using commercially available enzyme-linked immunosorbent assay (ELISA) kits (Cusabio, Houston, TX, USA) according to the manufacturer’s protocol. In brief, 25 µL each of the standard, control, and treated serum samples were added to the wells coated with anti-estradiol, anti-progesterone, anti-LH, and anti-FSH antibodies and incubated with enzyme conjugate for 2 h at room temperature. The wells were then washed with aqua dest. The wells were incubated with 100 µL of the substrate at room temperature. The reactions were stopped by adding a stop solution. Each sample was run in triplicate. The variations between the tests were less than 10%.

### 2.5. mRNA Expression of ER-α and ER-β

As shown in Table 1, purification, reverse transcription (RNA into cDNA), and quantitative real-time PCR were conducted in accordance with previous method on the uterus tissue [27].

### 2.6. Immunohistochemistry

The distributions of ERα and ERβ in the uterus of rats were evaluated using standard protocol by immunoCruz Rabbit ABC staining system kit (Santa Cruz, CA, USA) and immunoCruzGoat ABC staining system kit (Santa Cruz, CA, USA), respectively. Immunohistochemistry for these proteins used a similar principle and protocol but incubated with corresponding antibodies. All procedures for the immunohistochemical staining were according to Zaid et al., 2021 [27].

### 2.7. Statistical Analysis

The Statistical Package for Social Sciences (SPSS version 25.0) was used for statistical analysis. Data were presented as mean ± SEM. The normality distribution of the data was first examined using the Kolmogorov–Smirnov test. Then, data with normal distribution were analyzed using One-way Analysis of Variance (ANOVA) while not normally distributed data were analyzed using Mann–Whitney U-test. *p* < 0.05 was considered statistically significant.

## 3. Results

### 3.1. Body Weight and Uterine Weight

Table 2 and Figure 1a,b demonstrate the changes in body weight and uterine weight of the rats in all experimental groups. Although changes in the body weight gain were not statistically significant between the PS-MPs-exposed group with all other groups, six weeks of chronic exposure to polystyrene microplastics (PS-MPs) in the M group caused a 10% higher body weight change than the normal control rats (NC group). However, in the rats treated simultaneously with Kelulut honey (DM group), the increase in body weight was significantly reduced to the normal level compared with the normal control rats. Meanwhile, the changes in body weight of rats treated with Kelulut honey only (DC) were comparable to those of normal control rats.

A different pattern was observed in uterine weights. Following six weeks of PS-MPs exposure, there was a significant decrease in relative uterine weight of 22% in the M group compared with the normal control rats (NC group); however, concurrent treatment with Kelulut honey significantly reversed this effect. The relative uterine weight of the rats given only Kelulut honey (DC group) was comparable to that of the normal control rats (NC group), indicating that Kelulut honey had no adverse effect on uterine weight.

### 3.2. Histopathological Examination and Histomorphometry Analysis

Figure 2 shows a histopathological examination (HPE) of the uterus tissue of rats from all experimental groups. The quantitative results support this qualitative finding by histomorphometric analysis of the cellular components, including the height of the luminal epithelial cells, the thickness of the endometrium, myometrium, the height of endometrial epithelial glands, and the diameter of the endometrial glands.

As shown in Figure 2a(A1–A3), the normal control rats (NC group) had a normal histological appearance. The luminal epithelial was made up of tall pseudostratified epithelium located on a conspicuous basement membrane. Most luminal epithelial cells are cuboidal, thick, and contain a larger number of goblet cells with well-rounded nuclei. The endometrial appears healthy, with intact endometrial glands and high cellular content in the stroma. In addition, most of the endometrial glands in this group have a thick epithelial layer with a firm and circular shape without any contamination in the lumen. Moreover, the myometrium looked healthy and in a thick layer. The histological appearance of the uterus of rats treated with Kelulut honey only (group DC) was comparable to that of the normal control rats (NC group) (Figure 2d(D1–D3)).

As shown in Figure 2b(B1–B3), the histopathological findings in the uterus of the MPs-exposed rats (M group) differed from those of the normal control rats (NC group). In most cases, the thickness of the luminal epithelial was less and more compact, and some contained fewer goblet cells than the control rats (NC group). Furthermore, some of the endometrial glands appeared to be unhealthy due to contamination in the lumen, irregular shape, and a less organized and thin epithelial layer. In addition, the endometrium did not appear to be healthy due to oedema in the epithelial cells, with very little interstitial intracellular space between the stromal cells and a lower mitotic count. Moreover, most of the myometrial layer appeared smaller and thinner, with disintegrated muscle fibers.

Interestingly, these histopathological changes improved significantly and resembled the normal features of the normal control rats (NC group) when the PS-MPs-exposed rats were concurrently treated with Kelulut honey (DM group). For example, more goblet cells were found with an increase in the height of the luminal epithelial, with no oedema, and the stromal cells appeared to have increased in size; the interstitial gaps between the stromal cells appeared to have increased; the histology of the glandular epithelium resembled normal, and mitotic figures were visible. In addition, the myometrium was thick and almost normal (Figure 2c(C1–C3)).

Table 3 shows the results of histomorphometry of the uterus for all experimental groups. Overall, although histomorphometric changes (including luminal epithelium, endometrial thickness, myometrial layers, epithelial endometrial gland height, and endometrial gland diameter) were not statistically significant between the PS-MPs-exposed group and all other groups, a significant decrease in the MPs-exposed rats (M group) was observed compared with normal control rats (NC group), consistent with deleterious histological changes and uterine weight reduction. Interestingly, the concurrent treatment of rats with PS-MPs and Kelulut honey (DM group), significantly prevented the decrease in these parameters. In summary, the qualitative and quantitative results demonstrate that concurrent treatment of Kelulut honey in MPs-exposed rats (DM group) with Kelulut honey reduces the harmful changes in the uterus of rats due to microplastic exposure.

### 3.3. Gonadotropins Hormones (FSH and LH) and Sex Steroid Hormones (17β-Estradiol and Progesterone)

Figure 3 shows the levels of estradiol, progesterone, LH, and FSH and the hormones were analyzed to investigate the effects of PS-MPs and Kelulut honey exposure on reproductive hormone concentrations. The serum levels of FSH and LH have significantly reduced in the PS-MPs-exposed rats (M group) compared with the normal control group (NC group). It was found that concurrent treatment with Kelulut honey significantly prevented the reduction in LH levels in PS-MPs-exposed rats, which is comparable to the normal control group (NC group). However, this concurrent treatment resulted in a significant increase in serum FSH levels compared with all other groups. The FSH level of the rats treated with Kelulut honey only (DC group) was comparable to that of the NC group.

The 17β-estradiol (E2) and progesterone levels were significantly lower in PS-MPs-exposed rats (M group) compared with control rats (NC group). Concurrent treatment with Kelulut honey in PS-MPs-exposed rats (DM group) normalized these two hormone levels close to the normal control rats (NC group). These hormone levels were comparable between normal control rats (NC group) and rats treated with Kelulut honey alone (DC group).

### 3.4. ERα and ERβ Expression

The mRNA expression levels of ERα and ERβ were measured to correlate with their protein expression patterns in immunohistochemistry analyses (Figure 4). The expression of Erα in Figure 4a was substantially higher in the PS-MPs-exposed rats (M group) than in the normal control rats (NC group). The disruptive effect of PS-MPs on ERα expression was significantly inhibited with concurrent treatment of Kelulut honey (DM group). Meanwhile, mRNA ERα expression in the DC group was comparable with the NC group.

Figure 4b shows a similar pattern of ERβ mRNA expression with Erα mRNA expression. Compared with the normal control rats (NC group), the PS-MPs-exposed rats (M group) showed significantly higher levels of ERβ mRNA expression. The suppressive effect of PS-MPs on ERβ expression was significantly inhibited by concurrent treatment with Kelulut honey (DM group) where the expression is significantly close to that of the normal control rats (NC group). The ERβ expression levels of the rats treated with Kelulut honey alone (DC group) were also comparable to those of the normal control rats (NC group).

### 3.5. ERα and ERβ Protein Distribution

To analyze cell-specific changes in ERα and ERβ proteins, representative uterine tissue sections were stained using an immunohistochemical method. These proteins are distributed in the nuclei of uterine epithelial, myometrial, and stromal cells. Figure 5 shows the distribution of ERα in uterine tissues of all experimental groups. Compared with the normal control rats (NC group: B1, B2), a higher intensity of immunostaining was observed in PS-MPs-exposed rats (M group: C1, C2). Concurrent treatment with Kelulut honey (DM group: D1, D2) reduced the color intensity, which was comparable to that of DC group (E1, E2), however, having lower color intensity compared to NC group.

Figure 6 illustrates the differences in ERβ immunostaining intensity. Among all experimental groups, the M group had the highest intensity of immunostaining (C1, C2). However, lower intensity of immunostaining was observed after simultaneous treatment with Kelulut honey (DM group: D1, D2), however, the intensity of immunostaining was comparable in the NC and DC groups (B1, B2, and E1, E2, respectively). Figure 5(A1, A2) and Figure 6(A1, A2) shows blue color of immunostaining because no primary antibody (the negative control) was used. Only the blue hematoxylin counterstaining can be seen.

## 4. Discussion

Chronic exposure to numerous products containing microplastics (MPs) can lead to long-lasting changes in the reproductive system that may increase the risk of infertility. This study examined the harmful effects of polystyrene microplastics (PS-MPs) on the reproductive system of prepubertal animal models because the crucial time for a child’s development is prepuberty when the gonads grow rapidly before reaching maturity [30]. To our knowledge, this is the first work to systematically investigate how PS-MPs affect uterine function in the rat, including uterine morphology, hormone disruption, as well as gene and protein expression. Consequently, the possible protective abilities of Kelulut honey as a natural product against the disruptive effects of PS-MPs on the selected parameters were investigated. In this study, rats were treated from 28 to 70 days of age (6 weeks PS-MPs exposure), which corresponds to 2 and 15 years of age (puberty) in humans.

PS-MPs can ingest toxic substances and often contain additives, such as endocrine-disrupting chemicals (EDCs), that can harm organisms. Because PS-MPs can enter the uterus as foreign bodies and serve as carriers for EDCs to enter the human body, they can cause severe inflammation, oxidative stress, and even apoptosis. In addition, exposure to EDCs can increase the risk of developing obesity-related diseases [31]. In this study, we evaluated the effects of PS-MPs on the metabolic processes of rats and found that the body weight of PS-MPs-exposed rats was slightly increased but not significantly different from that of control rats. This finding, which was confirmed by previous studies, revealed that PS-MPs were not directly related to weight gain [23,27,32]. Based on toxicological studies, changes in the body and organ weight between untreated and treated animals may be valid indicators of the toxic effects of the test substances [33,34]. However, the disparity in body weight change findings remains unknown because various factors, including rat strain sensitivity and differences in dose, delivery method, age, and exposure time, may influence body weight. Thus, in this study, it was found that the body weights of rats did not change significantly but specific factors of PS-MPs (as mentioned above) affect metabolic pathways and promote the growth of fat cells [34], which can lead to an increase in body weight, but related mechanisms cannot be ignored. However, Kwon et al. found that treatment with plastic additives at high doses did not affect body weight in rats [35]. Therefore, further studies are needed to determine the effects of long-term treatment with PS-MPs on the body weight of rats.

The toxicological data from this investigation revealed that the effects of PS-MPs appear to be very specific to the uterus, as the weight loss only affects the uterus, and has no substantial effects on total body weight. These current findings are consistent with previous publications by Ashby et al., (2003), Liu et al., (2022), Zaid et al., (2015), and Zaid et al. (2021) [23,27,32,36]. Based on a toxicological study, the uterus is an organ that is sensitive to xeno-estrogen impacts [15,37,38]. When changes in organ weight of 10% or more are detected, the toxic effects on the organs are significant. A bioassay study revealed that the uterine weight of rats treated with PS-MPs was 22% less than that of control rats. This significant reduction in uterine weight corresponds with histological abnormalities, notably a decrease in the thickness of the luminal and interstitial space between the stromal cells, the presence of unhealthy glands, and a reduction in the thickness of the myometrial layer. Our findings are thus in line with previous studies [23,27,39]. Furthermore, the histological abnormalities reported here resembled those described in old rats with reproductive senescence (chronic anoestrus or persistent dioestrus) [38,40,41], such that the expression of estrogen-sensitive genes is altered in old rats, indicating a reduction in reproductive function [42].

MPs can disrupt uterine histology by increasing ROS generation in cells and tissues, resulting in oxidative stress. In addition, ROS causes apoptosis in gonadal tissues, resulting in tissue damage evidenced by histopathological alterations in reproductive tissues/gonads. MPs can also disrupt the ROS scavenger system and decrease the gene expression or activity of these enzymes [43]. Higher ROS levels, on the other hand, can deplete cellular ROS scavenger molecules as they are used to combat oxidative stress. It has been shown that MPs-induced ROS can lead to DNA damage and abnormalities in reproductive cells, resulting in reduced fertility [44]. In addition, signs of mitochondrial membrane potential disruption were observed in humans treated with PS-MPs as well (Caco-2), suggesting a universal mechanism that can lead to increased formation of ROS and toxicity in animals after exposure to MPs [45]. Moreover, oxidative stress at sustained high levels of ROS can damage DNA, lipids, and proteins, leading to cellular and tissue abnormalities. Therefore, increased mitochondrial dysfunction or ROS-induced cellular damage can be a likely explanation for the damage to reproductive tissue leading to lower fertility in animals.

Severe morphological abnormalities in the uterus of PS-MPs-exposed rats, including disruption of the normal structure of the luminal epithelium, the thickness of the endometrium, myometrium, and glandular epithelial cells, are associated with oxidative stress. These morphological changes can be due to the possible direct effects of EDCs (PS-MPs) on DNA, leading to altered gene expression. EDC-induced DNA damage has been documented in both in vivo and in vitro studies [46,47]. Interestingly, the disruptive effects of PS-MPs on uterine morphology were restored by concurrent treatment with Kelulut honey, which approximated normal histology. In addition to the carbohydrates, amino acids, proteins, organic acids, vitamins, and various phytochemicals contained in Kelulut honey, the improvement can be explained by the high content of natural antioxidants in honey [48].

The antioxidant effect of Kelulut honey is mainly due to the high presence of phenols (chlorogenic acid, p-coumaric acid, quercetin, epicatechin, rutin, catechin, and protocatechuic acid) [49], flavonoids, vitamins, and enzymes. Phenolic acids and flavonoids are the components of phenolic compounds and are responsible for preventing the formation of free radicals and inhibiting enzyme systems that produce free radicals. Phenolic compounds in Kelulut honey can also increase the concentration of biologically significant endogenous antioxidants and induce the expression of numerous genes involved in the production of enzymes that can prevent oxidative stress. In addition, Kelulut honey contains a high concentration of free radical scavengers that can minimize the imbalance between free radical formation and antioxidant levels, thereby reducing morphological abnormalities in rats. This finding was supported by Adenan et al., who found that Kelulut honey from *Trigona* sp. increased antioxidant defenses, elevated Nrf2 expression, and decreased lipid peroxidation and oxidative DNA damage [50]. In addition, Kelulut honey has been shown to reduce oxidative stress (a precursor of DNA damage) in lymphoblastoid cell lines (LCLs) through its free radical scavenging activities (phenolic and flavonoid chemicals) [48]. Furthermore, Kek et al. reported that Kelulut honey possesses higher phenolic content and has the highest antioxidant levels compared with Tualang and Gelam honey [17]. This high antioxidant content can counterbalance the free radicals generated by the chemical microplastic and restore the natural process in the uterus to normal.

In this study, we demonstrated that the levels of gonadotropins (FSH and LH) and sex steroid hormones (estradiol and progesterone) decreased in PS-MPs-exposed rats, suggesting that PS-MPs may affect the hypothalamic–pituitary–gonadal axis and hormone balance. Our findings are in agreement with the previous report by Rubin et al. [51] and Jin et al. [52]. Several studies have documented how MPs disrupt the rat estrous cycle [53]. The disruption of the estrous cycle in rats exposed to MPs can be caused by an alteration in the typical functioning of the hypothalamic–pituitary axis, which impairs the production of gonadotropin-releasing hormone and reduces the release of FSH and LH. As a result, it may have other disruptive effects on the development and functions of the follicles in the ovaries (the reason for the formation of cystic follicles and consequently the reduction in the formation of corpus luteum), causing the anovulation of the follicles and the interruption of the production of sex steroid hormones (E2 and progesterone) by the ovaries. Thus, the disruptive effect of PS -MPs on the hypothalamic–pituitary–gonadal axis can lead to abnormalities of ovarian and uterine functions and affect the normal secretion of gonadotropins and steroid hormones. Ijaz et al. also demonstrated that the impairment of reproduction caused by MPs due to the disruption of the normal function of GnRH [54] is essential for providing the primary hypothalamic signal for gonadotropin synthesis and secretion.

The increase in the percentage of the normal estrous cycle in PS-MPs-exposed rats should, theoretically, be related to the hypothalamic–pituitary axis, where restoration of gonadotropin hormone levels should be evident. Interestingly, treatment with Kelulut honey in PS-MPs-exposed rats almost restored the normal level of FSH, LH, progesterone, and E_2_. These observations were also confirmed by a previous study by Kamal et al. [55]. The improvement in gonadotropins and sex steroid hormones in this study can be explained by Kelulut honey’s phytochemical such as flavonoids and phenolic acids or bioactive compound content such as kaempferol and quercetin, which have antioxidant properties that help to restore hormone balances. In addition, a previous study suggested that Tualang honey can improve the disruption of hormone balance and the estrus cycle caused by bisphenol A in the uterus of rats [23]. Our research also supports the findings that Kelulut honey has higher antioxidant properties documented in three other studies showing that Kelulut honey increases SOD and GSH levels in testicular oxidative damage [56], increases SOD in osteoporosis rats [57], and increases catalase, SOD, and GSH peroxidase activities in PCOS rats [55].

This study also examined estrogen-responsive genes in utero to support the histology and oxidative stress findings, which showed that PS-MPs can affect uterine development, growth, and function by affecting the regulation of ERα and ERβ mRNA expression and protein distribution [23,58,59,60]. The estrogen receptor (ER) is a ligand-activated enhancer protein that is activated by the hormone estrogen (17-oestradiol) and can control gene transcription via estrogen-responsive elements [23]. Unfortunately, it can also be triggered by other substances, such as EDCs, that leak from PS-MPs [61,62]. Compared with exogenous and/or synthetic estrogen, the binding affinity of endogenous estrogen to ERα and ERβ is lower [63], but the transactivation of both receptors via the estrogen-responsive element is comparable (ERE). However, their roles in transcriptional activity, which are highly dependent on the ligands and their responsive components, are different [64]. ERβ functions as a transdominant repressor that inhibits ERα transcriptional activity [65]. The ability of ERβ to form heterodimers with ERα, which in turn govern the actions of estrogen receptors, causes inhibitory effects [66]. In contrast to ERα, Erβ thus has a disruptive effect on ERα rather than the typical uterotrophic effects.

In the present study, the upregulation of ERα and ERβ in PS-MPs-exposed rats may be due to uterine dysfunction in addition to the effects of lipid peroxidation. EDCs have an estrogenic effect (EA) and are also known to be chemicals that can mimic or counteract the effects of naturally occurring estrogens. These hormone disruptors can therefore interfere with the normal action of estrogen and other hormones in the body by blocking or mimicking them, thereby disrupting the body’s hormonal balance [67]. Estrogen is essential for sexual differentiation. Because estrogen disruptors (EDCs) have a structural similarity to natural estrogen, they can bind and activate estrogen receptors and exhibit a similar response even in the absence of natural estrogen, which can cause oxidative stress, histological abnormalities, and hormonal imbalance. This may explain our current findings of significant upregulation of ERα and ERβ in PS-MPs-exposed rats (M group) compared with control rats. These results are consistent with a previous study by Roy et al. [67]. Furthermore, using quantitative RT-PCR and immunohistochemistry, our study discovered that the effects on ERα and ERβ expression levels were different from those shown by estradiol hormone levels. This showed that EDCs in PS-MPs can reduce natural estradiol hormone secretion, but can also structurally resemble estrogen, increasing protein expression and distribution and thus disrupting normal uterine function.

Interestingly, this study found an improvement in mRNA expression and protein distribution of Erα and ERβ in PS-MPs-exposed rats treated with Kelulut honey. This can be explained by the fact that Kelulut honey is high in antioxidants, including flavonoids, quercetin, carotenoids, and phenolic compounds. Due to their structural similarity to 17-oestradiol, the two main naturally occurring flavonols, quercetin and kaempferol, may have weak estrogenic effects [68,69]. According to data from previous studies, antioxidants have a well-known function in oxidative stress/damage linked to various diseases. Antioxidants not only scavenge free radicals but also modulate signal transduction pathways that are affected by free radicals during oxidative stress and are responsible for cellular responses to various diseases [70,71,72]. In addition to its potential to scavenge oxidants and free radicals, the physiologically active estrogen-like substance in Kelulut honey reduced uterine atrophy. Furthermore, previous studies have shown that the possible competitive binding of these chemicals led to an improvement in the function of the hypothalamic–pituitary axis in BPA-exposed rats [22].

The components responsible for the high antioxidant properties of Kelulut honey probably include phenolic acids, flavonoids, vitamins, and enzymes. Honey also contains many free radical scavengers that can help balance free radical production and antioxidant levels [48]. In addition, the flavonoids in Kelulut honey contribute to the protective effect against the toxicity of PS-MPs by scavenging and reducing ROS in cells. In fact, this medicinal plant also contains vitamins (C and E), and non-enzymatic antioxidants that play a crucial role in removing free radicals from body tissues [51,73,74] and may help reduce reproductive toxicity due to exposure to PS-MPs.

## 5. Conclusions

In conclusion, the uterus of prepubertal rats underwent disruptive changes in organ and body weight, hormone profile, histology, mRNA expression, and protein distribution due to PS-MPs exposure. Interestingly, the phytochemical properties of Kelulut honey can reduce the uterine toxicity of PS-MPs. Our results show that Kelulut honey improved oxidative stress, hormonal profile, and histological and sex steroid receptor expression in PS-MPs-exposed rats. Kelulut honey has the potential to be used in medicine as it contains phenols that can increase antioxidant levels and reduce oxidative stress-related diseases. In view of this, Kelulut honey can be helpful as a health supplement to reduce or prevent the risk of severe reproductive infertility caused by long-term PS-MPs exposure, especially during puberty. However, further studies are needed to discover the mechanisms of the bioactive components in Kelulut honey that contribute to this protective effect on the female reproductive system.

## Figures and Tables

**Figure 1 toxics-11-00324-f001:**
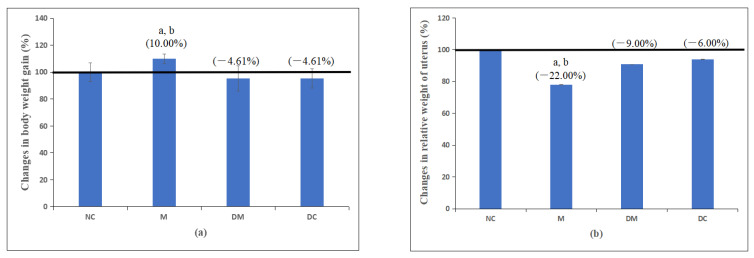
(**a**) Changes in body weight gain and (**b**) uteri-relative weight in all experimental groups. NC: normal control group; M: microplastics group (PS-MPs 2.5 mg/kg); DM: Kelulut honey group (PS-MPs 2.5 mg/kg + KH 1200 mg/kg); and DC: Kelulut honey control group (KH 1200 mg/kg). Data are expressed as mean ± SEM. ^a^
*p* < 0.05 vs. NC. ^b^ *p* < 0.05 vs. DM.

**Figure 2 toxics-11-00324-f002:**
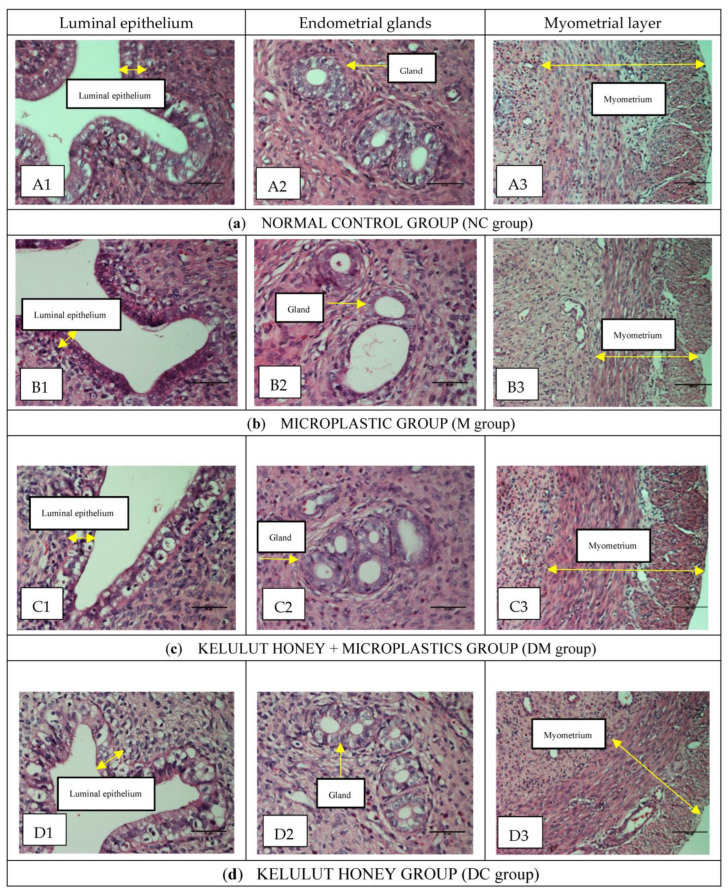
Histological findings of the luminal epithelium (H and E, ×400), endometrial glands (H and E, ×400), endometrial stroma, and myometrium (H and E, ×200) of uterine from all experimental groups. NC: Normal control group; M: microplastics group (PS-MPs 2.5 mg/kg); DM: Kelulut honey group (PS-MPs 2.5 mg/kg + KH 1200 mg/kg); and DC: Kelulut honey control group (KH 1200 mg/kg).

**Figure 3 toxics-11-00324-f003:**
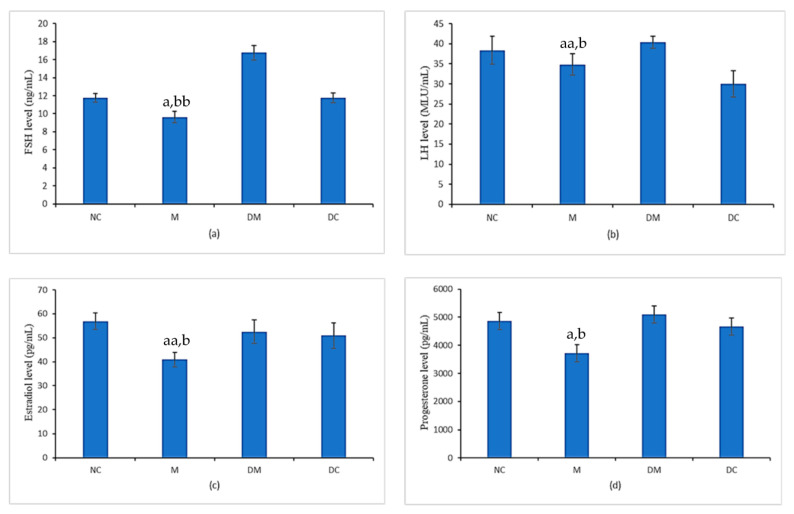
Level of (**a**) 17β-estradiol ($E_{2})$, (**b**) progesterone, (**c**) FSH and (**d**) LH for all experimental groups. NC: normal control group; M: microplastics group (PS-MPs 2.5 mg/kg); DM: Kelulut honey group (PS-MPs 2.5 mg/kg + KH 1200 mg/kg); and DC: Kelulut honey control group (KH 1200 mg/kg). Data are expressed as mean ± SEM. ^a^
*p* < 0.05 vs. NC. ^aa^
*p* < 0.01 vs. NC. ^b^ *p* < 0.05 vs. DM. ^bb^ *p* < 0.01 vs. DM.

**Figure 4 toxics-11-00324-f004:**
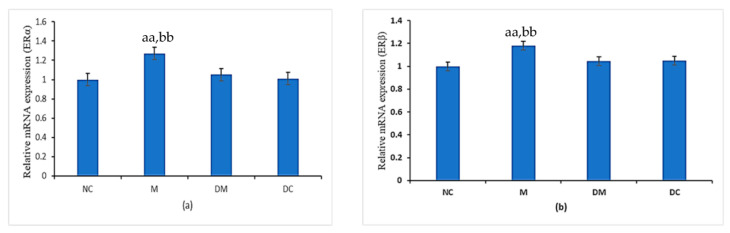
(**a**) The ERα gene’s relative quantitative expression in all experiment groups. (**b**) The ERβ gene’s relative quantitative expression in all experiment groups. NC: normal control group; M: microplastics group (PS-MPs 2.5 mg/kg); DM: Kelulut honey group (PS-MPs 2.5 mg/kg + KH 1200 mg/kg); and DC: Kelulut honey control group (KH 1200 mg/kg). Data are expressed as Mean ± SEM. ^aa^ *p* < 0.01 vs. NC and ^bb^ *p* < 0.01 vs. DM.

**Figure 5 toxics-11-00324-f005:**
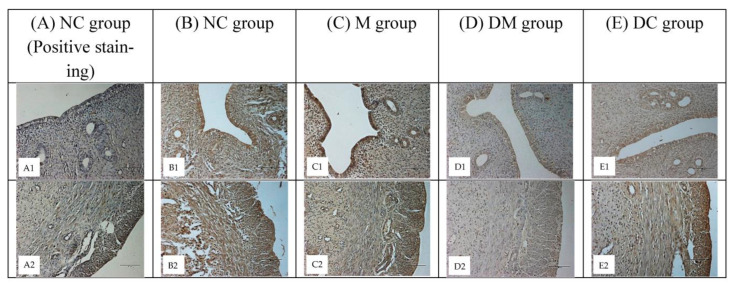
Immunohistological localization of ERα in uterine section in all experimental groups (×200). NC: normal control group; M: microplastics group (PS-MPs 2.5 mg/kg); DM: Kelulut honey group (PS-MPs 2.5 mg/kg + KH 1200 mg/kg); and DC: Kelulut honey control group (KH 1200 mg/kg).

**Figure 6 toxics-11-00324-f006:**
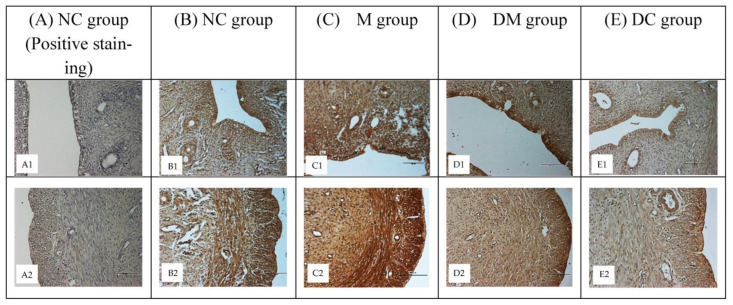
Immunohistological of ERβ in the uterine section of all experimental groups (×200). NC: normal control group; M: microplastics group (PS-MPs 2.5 mg/kg); DM: Kelulut honey group (PS-MPs 2.5 mg/kg + KH 1200 mg/kg); and DC: Kelulut honey control group (KH 1200 mg/kg).

**Table 1 toxics-11-00324-t001:** Reverse and forward primers of ERα and Erβ for Taqman PCR analysis.

Gene	Reverse Primer/Forward Primer	References (Accession Number)
Β-actin	5′-GCCGGGACCTGACTGACTAC-3′ 5′-TTCTCCTTAATGTCACGCACGAT-3′	Kuipter et al., 1996 [28] (U57439)
ERα	5′-GCAGGTCATAGAGAGGCACGA-3′ 5′-AAGCTGGCCTGACTCTGCAG-3′	Spreafico et al., 1992 [29] (X61098)
ERβ	5′-GGAGATACCACTCTTCGCAATC-3′ 5′-CTCTGTGTGAAGGCCATGAT-3′	Kuipter et al., 1996 [28] (U57439)

**Table 2 toxics-11-00324-t002:** Body weight gain, changes in body weight, uterine wet weight, and relative weight of uterus in all experimental groups.

Group	Body Weight Gain (g)	Changes in Body Weight Gain (%)	Uterine Wet Weight (mg)	The Relative Weight of the Uterus (Wet Weight/Body Weight)
NC	144.5 ± 4.13	60.35 ± 11.37	333 ± 12.02	1.84 ± 0.11
M	151.5 ± 3.29 ^a,b^	62.70 ± 17.74 ^a,b^	260 ± 2.00 ^a,b^	1.46 ± 0.07 ^a,b^
DM	141.7 ± 5.59	59.18 ± 18.20	303 ± 3.09	1.59 ± 0.04
DC	141.5 ± 8.33	59.17 ± 15.56	314 ± 14.59	1.72 ± 0.10

Data are expressed as Mean ± SEM. NC: normal control group. M: positive control group (2.5 mg/kg PS-MPs). DM: Kelulut honey group (1000 mg/kg Kelulut honey + 2.5 mg/kg PS-MPs). DC: Kelulut honey control group (1000 mg/kg Kelulut honey). ^a^ *p* < 0.05 vs. NC. ^b^
*p* < 0.05 vs. DM.

**Table 3 toxics-11-00324-t003:** Histomorphometry analysis of the uterus in all experimental groups.

Group	Height of Luminal Epithelial Cell (µm)	The Thickness of the Endometrium (µm)	The Thickness of Myometrium (µm)	Height of Epithelial Endometrial Glands (µm)	Diameter of Endometrial Glands (µm)
NC	29.12 ± 2.38	643.55 ± 15.83	343.4 ± 17.10	19.36 ± 1.29	71.11 ± 4.58
M	23.71 ± 1.013 ^a,b^	562.65 ± 12.51 ^a,b^	290.96 ± 15.93 ^a,b^	15.66 ± 0.22 ^a,b^	68.43 ± 9.28 ^a,b^
DM	35.21 ± 2.351	581.84 ± 24.69	378.89 ± 17.43	17.25 ± 1.60	65.79 ± 2.35
DC	31.90 ± 1.462	572.48 ± 23.58	322.83 ± 16.06	20.72 ± 1.40	76.26 ± 2.49

Data are expressed as Mean ± SEM. NC: normal control group. M: positive control group (2.5 mg/kg PS-MPs). DM: Kelulut honey group (1000 mg/kg Kelulut honey + 2.5 mg/kg PS-MPs). DC: Kelulut honey control group (1000 mg/kg Kelulut honey). ^a^ *p* < 0.05 vs. NC. ^b^
*p* < 0.05 vs. DM.

## Data Availability

Not applicable.

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
