# Peer review of "Protective Role of Kelulut Honey against Toxicity Effects of Polystyrene Microplastics on Morphology, Hormones, and Sex Steroid Receptor Expression in the Uterus of Rats"

_toxics, 2023, doi:10.3390/toxics11040324_

Round 1

Reviewer 1 Report

Please revise the following. Some of these comments are major requirements, without which the reviewer cannot evaluate the authors conclusions.

1.     Typo in Section 2.2 heading  - "experimental"

2.     Table 1 – please state explicitly if the p value from statistical tests between all other groups was not significant. Also, what statistical test was used? How was this determined as the right test to use?

3.     Figure 1: Depicting data as an individual value chart will be more useful than bar graphs, especially since we can’t really see the error bars in Figure 1b.

4.     Figure 2 legend: Magnification x200 or x400 does not provide any useful information. Please clearly state the actual length measurement (micrometers) of the scalebar in each type of image.

5.     Label for Table 2 is incorrect, it says Table 1 again. The heading/description is also wrong.

6.     Table 2 – please state explicitly if the p value from statistical tests between all other groups was not significant. Also, what statistical test was used?

7.     Figure 3: Labels denoting the p values are missing from the figure. This is a major correction required, as this reviewer is unable to evaluate the authors’ conclusions without this information. If the p value was not significant, please mention "n.s.". Also, which groups were compared?  

8.     Figure 4: Labels denoting the p values are missing from the figure. This is a major correction required, as this reviewer is unable to evaluate the authors’ conclusions without this information. If the p value was not significant, please mention "n.s.". Also, which groups were compared?  

9.     Figure 4: Please explicitly state in the Methods section the primers sequences used for qPCR of the ERalpha and ERbeta transcripts. Also, both these receptors express multiple transcriptional variants, how did you choose this particular variant? Please provide rationale on not measuring other variants. Alternatively, please measure other variants known to be expressed in the rat uterus.  Please also state the reference gene used for gene expression normalization.

10.  Results section 3.4: Please explicitly state the tissue source (uterus?) for the gene expression studies here, and in the Methods section.

11.  Figure 5: There is lot more blue staining in the NC Positive Staining panel that others, please explain – positive staining should be brown, not blue. Also, the white balance is very different from other panels, please normalize. Higher magnification images are an absolute requirement to understand the cellular localization of ERalpha – without higher magnification, this is critical information lost. While it may not be pertinent to the conclusions of this paper, further studies looking into mechanisms of Kelulut honey will greatly benefit from high magnification images, which the authors already possess.

12.  Figure 6: There is lot more blue staining in the NC Positive Staining panel that others, please explain – positive staining should be brown, not blue. Also, the white balance is very different from other panels, please normalize. Higher magnification images will be helpful.

13.  What efforts were taken to make sure there is no cross-reactivity of ERalpha antibodies with ERbeta and viceversa? Please comment on antibody choice and if both ERalpha and ERbeta are known to be expressed and estrogen-responsive (nuclear localized) in normal rat uterine tissue (provide references from previous studies).

Author Response

Dear Editors/Reviewers, 

Thank you very much for our valuable comments and suggestions. Attached here - the point-to point responses to the comments. 

Thank you

Reviewer 2 Report

Reviewer’s Comment

This article is well written and the experimental protocols were followed perfectly, design of experiments and approval were also done well. The authors have achieved only 50% of their goal because the experiment with toxicity PS-MP is successful but the role of Kelulut honey is not scientifically proven or justified.

Effects of PS-MP on organ  and body weight, hormone profile, histology, mRNA expression, and protein distribution were explained neatly with statistical analysis but the title says “
Protective role of Kelulut honey against disruptive effects of PS-MP……”  here protective role of the honey was not well explained or demonstrated.

A general reader may have the following question,

1.     What phytochemical or active compound from the honey is protecting the uterine health?

2.     How honey can play role on the sex steroids or ER alpha, ER beta expressions or other metabolic activities like obesity or organ weight gain?

3.     Any random animal histology of any organ may show differences and may not be the same, it may be control or treated, how will you justify its because of honey?

Hence the discussion and conclusion section can be concentrated more on role of honey on these parameters with previous studies or propose a hypothesis with your findings.

Line no 109 has a spelling mistake “2.2. Animal model and eperimental design”

Author Response

Dear Editors/Reviewers,

Thank you very much for your valuable comments and suggestions. Attached here-the point to point responses to the comments

Thank you

Round 2

Reviewer 1 Report

Response from authors is satisfactory.